# Antifungal Evaluation and Molecular Docking Studies of *Olea europaea* Leaf Extract, *Thymus vulgaris* and *Boswellia carteri* Essential Oil as Prospective Fungal Inhibitor Candidates

**DOI:** 10.3390/molecules26206118

**Published:** 2021-10-10

**Authors:** Hanaa S. Omar, Soheir N. Abd El-Rahman, Sheikha M. AlGhannam, Nour El-Houda A. Reyad, Mohamed S. Sedeek

**Affiliations:** 1Department of Genetics, Faculty of Agriculture, Cairo University, Giza 12613, Egypt; 2GMO Laboratory, Faculty of Agriculture, Cairo University, Research Park, CURP, Giza 12613, Egypt; 3Crops Technology Research Department, Food Technology Research Institute, Agricultural Research Center, Giza 12619, Egypt; 4Department of Chemistry, College of Science, Imam Abdulrahman Bin Faisal University, P.O. Box 1982, Dammam 31441, Saudi Arabia; sm_ghannam@yahoo.com; 5Plant Pathology Department, Faculty of Agriculture, Cairo University, Giza 12613, Egypt; nourelhoudareyad53@gmail.com; 6Pharmacognosy Department, Faculty of Pharmacy, Cairo University, Giza 11562, Egypt; mohamed.sedeek@pharma.cu.edu.eg

**Keywords:** *Olea europaea* leaf extracts, *Fusarium oxysporum*, *Thymus vulgaris*, internal transcribed spacer (ITS), molecular docking

## Abstract

**Background:** The present study investigated the antifungal activity and mode of action of four *Olea europaea* leaf extracts, *Thymus vulgaris* essential oil (EO), and *Boswellia carteri* EO against *Fusarium oxysporum*. **Methods:**
*Fusarium oxysporum lactucae* was detected with the internal transcribed spacer (ITS) region. The chemical compositions of chloroform and dichloromethane extracts of *O. europaea* leaves and *T. vulgaris* EO were analyzed using GC-MS analysis. In addition, a molecular docking analysis was used to identify the expected ligands of these extracts against eleven *F. oxysporum* proteins. **Results:** The nucleotide sequence of the *F. oxysporum* lactucae isolate was deposited in GenBank with Accession No. MT249304.1. The *T. vulgaris* EO, chloroform, dichloromethane and ethanol efficiently inhibited the growth at concentrations of 75.5 and 37.75 mg/mL, whereas ethyl acetate, and *B. carteri* EO did not exhibit antifungal activity. The GC-MS analysis revealed that the major and most vital compounds of the *T. vulgaris* EO, chloroform, and dichloromethane were thymol, carvacrol, tetratriacontane, and palmitic acid. Moreover, molecular modeling revealed the activity of these compounds against *F. oxysporum*. **Conclusions:** Chloroform, dichloromethane and ethanol, olive leaf extract, and *T. vulgaris* EO showed a strong effect against *F. oxysporum*. Consequently, this represents an appropriate natural source of biological compounds for use in healthcare. In addition, homology modeling and docking analysis are the best analyses for clarifying the mechanisms of antifungal activity.

## 1. Introduction

*Lactuca sativa* (*Lettuce*) is one of the most commonly used crops in the leafy vegetable group, and it belongs to the family of *Asteraceae* [1]. Lettuce is a vital dietary vegetable, and it has some health benefits as a source of vitamin C, phenolic compounds, and fiber [2]. Overall, lettuce is susceptible to various diseases, such as viral and fungal infections, including powdery mildew, fungus, and bacterial wilt.

In this respect, the search for potent and selective inhibitors has received particular attention because of the increase in antifungal resistance, which has become one of the greatest challenges for global health, food security, and development. In particular, in the agricultural field, *F. oxysporum* forma sp. *Lettuce* (*F.O.L.*) represents the most serious agent of all pathologies involved in lettuce cultivation, especially in North Africa [3]. *F. oxysporum* forma sp. *Lettuce* (*F.O.L.*) is the most important and common fungal pathogen in lettuce plants, and it causes wilting and yield reduction. It remains with a specific host; it increases the yellowing of the leaves and wilting, and it affects the vascular system of the lettuce plant [4]. Unfortunately, to this day, no curative treatment for this fungus exists, except for some limited approaches, such as the disinfection of the soil or the propagation and use of resistant varieties, which remain the first approaches to reducing the impact of this disease [5].

Synthetic fungicides are considered successful tools, and there is no doubt that they have been used for many years in traditional agricultural systems. However, the use of synthetic fungicides has now been reduced because of their adverse effects on humans and the environment. The need to look for and evaluate alternative solutions that are environmentally friendly, available, and affordable for smallholder farmers is important for these crop protection challenges [6].

Nowadays, much attention is required in order to study plant-based ingredients and essential oils for their broad range of biological activities, such as their antiviral, anti-inflammatory, antifungal, and antibacterial properties [7,8]. Essential oils (EOs) are considered essential sources of biologically active compounds, i.e., fungicidal, insecticidal, antibacterial, nematocidal, and herbicidal compounds [8,9]. *T. vulgaris* includes several species of aromatic herbaceous plants. Moreover, its oil can be used externally as an antiseptic for fungal infections. Thymol is part of a naturally occurring class of compounds identified as biocides, which have strong antimicrobial traits when used alone or with other biocides, such as carvacrol [10]. *O. europaea* leaf extracts contain phenolic compounds that are effective against bacteria, mycoplasma, and yeasts [11]. Phenolic composites are recognized for their inhibitory activity against bacteria, fungi, and viruses [12]. An *O. europaea* leaf extract was found to inhibit fungi and Gram-positive bacteria [13].

Recently, the antifungal mechanisms of action of EOs were identified by determining the ergosterol content of the plasma membranes of fungi [14]. Ergosterol is the main sterol component of the fungal cell membrane, and it is responsible for cell function and integrity. Ergosterol is found in almost all fungi, and it is widely used as an indicator of fungal biomass. Therefore, the essential oils of conventionally studied plants were identified as essential instruments in the formulation of plant-based preservatives against aflatoxin and mold contamination, and also as protection against the destructive effects of free radicals [15].

On the other hand, due to inadequate information on the pathogenesis of the *F. oxysporum* fungus, several computational methods can be applied to better explain its mechanism of action. In this context, molecular docking analysis remains one of the most important tools, as it can provide atomistic insight into molecular recognition by predicting the ability of a molecule to bind to the active site of a protein. In our case, eleven *F. oxysporum* proteins were determined as targets for our docking studies. *F. oxysporum* (RHO1) has an essential role in maintaining the hyphal architecture and virulence of Fusarium spp., and it is also responsible for the regulation of the post-translational activity of glucan synthase, which inhibits its recognition by the host. *F. oxysporum* (XlnR) is responsible for the regulation of plant-cell-wall-degrading enzyme expression. Moreover, Fmk1 from *F. oxysporum* (MAPK) is a downstream transcription factor and has a vital role in causing invasive hyphal growth and plant infection [16,17].

No previous experiments have studied the effects of chloroform and dichloromethane extracts of *O. europaea* leaves or their modes of action against *F. oxysporum* forma species *lactucae* through molecular docking analysis. Therefore, this study aims to isolate and identify the purified *F. oxysporum* forma species *lactucae* using the ITS sequence of the conserved ribosomal DNA. Additionally, a preliminary screening and an evaluation of the phytochemical composition of the antifungal activities of some plant extracts, i.e., *T. vulgaris*, *Boswellia frankincense*, and *O. europaea* leaves, were used to test the effectiveness against the isolated *F*. oxysporum forma species *lactucae*. Moreover, a molecular docking analysis was performed for the expected antifungal ligands of these extracts in homology models that were constructed for the eleven *F. oxysporum* proteins in order to gain better insight into the ligand–protein binding interactions that prevent the infection process of Fusarium.

## 2. Results

### 2.1. Morphological Characterization, Isolation, and Pathogenicity Tests

The results of the morphological characterization and isolation of *F. oxysporum* f. sp. *lactucae* are presented in Figure 1a,b. The results reveal that the lettuce plants showed wilting symptoms as the result of a single, pure colony in potato dextrose agar (PDA) media. Moreover, the chlamydospores seemed to have an elliptical and spherical shape and were established only in a short, peripheral chain that reached 19.7 − 19.2 × 21.3 − 19.9 μm. A *F. oxysporum* isolate was able to infect the lettuce plants. It caused a disease incidence of 100% and a disease severity of 94.44%, but the control plants remained healthy. The initial symptoms appeared as yellowing on one side of the outer leaves, which then turned brown (Figure 1a). As the disease progressed, the plants wilted and died. After uprooting the infected plants from the soil and making a longitudinal section of the roots, discoloration could be noticed in the vascular area of the crown (Figure 1b). Based on the morphological characterization and isolation, the results reveal that the characteristics correspond to *F. oxysporum* infection.

### 2.2. Molecular Characterization and Phylogenetic Tree of the Purified F. oxysporum Isolate

The molecular identification and phylogenetic analyses of the purified *F. oxysporum lactucae* isolate were carried out (Figure 2a,b). The PCR product of the ITS sequence was detected at 650 bp, as shown in Figure 2a. The sequence analysis of the fungal isolate showed 100% similarity to the *F. oxysporum* sp. *lactucae* ITS sequence, and it was deposited in the Gene Bank under the accession number MT249304.1. In addition, the phylogenetic analysis confirmed that it had the highest similarity to the *F. oxysporum lactucae* isolate Lux (MT249304.1). The results of the phylogenetic tree analysis revealed that the closeness of the genetic similarity between the studied *F. oxysporum* isolate and others from around the world was mainly with strains of *F. oxysporum* with different accession numbers (MH855643.1, MH855398.1, DQ016234.1, MH855101.1, MH321792.1, JQ219941.1, and LC507102.1) in the database of the GenBank (Figure 2b). Therefore, *F. oxysporum* was recognized as the causal mediator of the Fusarium wilt in lettuce in Egypt.

### 2.3. Biological Activity

The antifungal properties of the ethanol, dichloromethane, chloroform, and ethyl acetate from *O. europaea* leaf extracts, and the oils of *T. Vulgaris* and *B.*
*carteri* against *F. oxysporum* f. sp. *lactucae* were evaluated in this investigation, as shown in Figure 3 and Figure 4 and Table 1. The results revealed that the *T. Vulgaris* essential oil and the dichloromethane, chloroform and ethanol extracts of O. europaea leaf extract leaves efficiently inhibited the mycelial growth of the *F. oxysporum* f. sp. *lactuca* compared to control treatment, whereas the ethyl acetate extracts did not exhibit antifungal effects at the tested concentration. The data illustrated in Figure 4 indicate that all the tested plant extracts except ethyl acetate extract significantly decreased the mycelial growth of the pathogenic fungi compared to the control treatment. The oil of T. Vulgaris showed the maximum inhibition percentage (94.11) at the tested concentration (75.5 mg/mL), against the fungus. Moreover, the chloroform, dichloromethane and ethanol extracts exhibited the maximum inhibition (67.058, 64.71 and 50.59 mm, respectively) at the tested concentration (75.5 mg/mL).

### 2.4. GC-MS Analysis

A post-silylation GC-MS analysis was employed to profile the primary metabolites. The analysis of the silylated dichloromethane and chloroform extracts *O. europaea* leaves led to the detection of 39 and 54 metabolites, respectively (Table 2 and Table 3). The identified components belonged to different classes, such as sugars, organic and amino acids, fatty acids, flavonoids, and low-molecular-weight or nonpolar secondary metabolites that were exemplified in alkaloids and steroids.

The separated oil was yellow with a spicy aromatic odor. The yield was 2.7% (*w*/*w*), and the refractive index was 1.4894. Twenty-four components were identified in the *T. vulgaris* essential oil at different percentages (Table 4). The GC-MS analysis of the oil revealed that the major components were thymol (41.85%), o-cymene (11.76%), gamma-terpinene (10.85), and carvacrol (3.61%). These four significant constituents represented 68.07% of all components.

### 2.5. Effects of Plant Phytochemical Extracts on the Virulence Proteins of F. oxysporum According to Docking Analysis

The eleven *F. oxysporum* proteins that were studied in relation to the active compounds present in *T. vulgaris*, *B. carteri* essential oil, and the *O.europaea* leaf extracts were docked as shown in Table 5, Figure 5 and Figure 6. This study was conducted to identify the ligands expected to block their activities and, hence, to gain a better clarification of their mode of action in controlling the pathogenicity of *F. oxysporum* f. sp. *lactucae*. These eleven vital proteins, i.e., AreA, MeaB, Fmk1, Ste7, Set12, Sge1, Xin R, Hog1, PacC, Mkk12, and Rho1, which have vital virulence pathways, were modeled. These proteins were the most important candidates in the virulence of *F. oxysporum*, as shown in Figure 4. In these connections, carvacrol, α-thujene, and thymol compounds bound with the active sites of these proteins with a binding affinity that ranged from −4.1 to −6.9 kcal/mol. Moreover, the carvacrol ligand had the highest score and bound with Hog1 (A) and PacC (B), while α–Thujene and thymol bound with Mkk12 (C) and Rho1 (D), respectively, as shown in Figure 5. It is worth mentioning that all of these compounds were present in the *T. vulgaris* plant extract. On the other hand, the nonacosane and tetratriacontane ligands bound with the FMK1, SET7, SEG1, and Rho1 proteins with a binding energy that varied from −5.2 to −5.9 kcal/mol. However, these compounds were found in chloroform extract of *O. europaea* leaves (Table 5). Hexadecenoic acid, palmitic acid, tetratetracontane, and Stigmast-5-en-3-ol (3á,24S) from the dichloromethane extract of the *O. europaea* leaves bound with the FMK1, SET7, SEG1, and Rho1 proteins with a binding energy ranging from −4.9 to −5.2 kcal/mol (Table 5). The model showed the docking between the virulence proteins of *F. oxysporum* and the ligands from the studied and tested plant extracts in order to explore the mechanisms of the binding of the selected proteins in an attempt to understand their role in the inhibition of the *F. oxysporum* pathogen.

## 3. Discussion

*Fusarium oxysporum* is a huge complex of species of plant and human pathogens that attack a wide array of species in a host-specific manner. Fusarium is a plant disease that exists in the soil. It penetrates into plants and causes losses crop yield and production [19]. Therefore, in this study, experiments were designed to isolate and identify *F. oxysporum* sp. *lactucae*. In addition, the biological activities of *T. vulgaris*, *B. carteri* and *O. europaea* leaves against the studied and isolated *F*. oxysporum were evaluated. Then, the investigation of their modes of action in preventing the development of the Fusarium infection process and in controlling the disease was assessed by using molecular docking analysis. As a perfect, sensitive, fast, and specific means of fungal identification and detection, many authors have established molecular methods as an alternative approach to the conventional procedures used in fungal identification. The internal transcribed spacer (ITS) region of the ribosomal DNA is highly variable within the genus Fusarium. In addition, the use of polymerase chain reaction (PCR) with primers targeted to this region for the detection and identification of *Fusarium* species with molecular methods was summarized by [20,21]. In this investigation, based on the morphological, cultural, pathogenic, and molecular results, the fungus was identified using the IT’S sequence of the ribosomal DNA, as *F. oxysporum* has the accession number MT249304.1. These results are in agreement with those described by [22], as they used the same method of using the ITS sequence of the ribosomal DNA to identify the differences among the species of the genus *Fusarium*. The results of the PCR identification of the ITS sequence successfully categorized the studied fungus as *F. oxysporum*; the fungus was not capable of infecting non-lettuce hosts. All of these data agree with the findings of [23]. They stated that *Fusarium* yielded isolates from lettuce plants that were particularly virulent in lettuce hosts. Therefore, it could be concluded that the fungus in the experiment was *F. oxysporum* f. sp. *lactucae*. According to the available research and our knowledge, Fusarium wilt in lettuce caused by *F. oxysporum* f. sp. *lactucae* was not previously documented in Egypt. Further surveys and in-depth investigations should be considered.

In this respect, the antifungal potentials of ethanol, dichloromethane, chloroform, and ethyl acetate extracts from *O. europaea* leaves, as well as those of the oil of *T. vulgaris,* were checked with respect to *F. oxysporum*. The results of the preliminary screening and evaluation revealed that the *T. vulgaris* essential oil and dichloromethane chloroform and ethanol extracts from *O. europaea* leaves efficiently inhibited the growth of the tested fungus with variable potency, whereas the ethyl acetate extract did not exhibit an antifungal effect at the tested concentration (75.5, 37.75 and 18. 875 mg/mL). These results are in line with those of other studies in which phenols extracted from *O. europaea* leaves showed antifungal activity [24]. In this respect, *T. vulgaris* has antimicrobial potential against pathogenic microorganisms [25]. The effectiveness of thyme EO against food-related bacteria and fungi was tested. The synergistic, antagonistic, and additive effects of the components of EOs require further research in order to elucidate the mechanisms underlying their biological activity and to access new and natural antiseptics that are applicable to the pharmaceutical and food industries [26].

The dichloromethane and chloroform extracts from *O.europaea* leaves effectively inhibited *F. oxysporum* f. sp. *lactucae* with zones of inhibition of 34.83 and 23.25 mm, respectively. The extracts under investigation were potent and natural antifungal drugs, which is in agreement with the studies of [27,28]. Metabolite profiling provided insights into the mediation of the metabolites in *O.europaea* leaf extracts for their effects as an initial step in establishing an assessment of the quality of the extracts.

In the current study, *T. vulgaris* essential oil significantly inhibited *F. oxysporum* f. sp. *lactucae* with a maximum inhibition percentage (94.11) at the tested concentration (75.5 mg/mL. The results of our research are in agreement with those of prior studies that showed the powerful antifungal activity of *T. vulgaris* oil [29]. Our results confirm that *T. vulgaris* can be used as a potent natural agent against foodborne pathogens and in the protection of valuable crops [30]. The principal components, thymol and carvacrol, play a vital role in the antifungal activity of the oil [31]. Therefore, they can serve as markers for the *T. vulgaris* essential oil.

The potent antifungal activity of the extracts and essential oil under investigation is promising in response to previous studies that encouraged natural drugs as pesticidal, antimicrobial, and food-preservative alternatives to chemical agents [32,33].

Computational modeling could be used to determine exceptional information in order to understand the mechanisms of the modes of action of the antifungal molecules that inhibit the fungal infection process. The molecular docking approach was used to predict the molecules that could bind specifically to the protein active sites that are responsible for the fungal infection process [34]. In the present study, the docking of the active molecules of the studied plant extracts with eleven essential proteins involved in the development pathway of *F. oxysporum* was evaluated. In this respect, the results of the molecular docking analysis showed that the carvacrol, α-thujene, and thymol compounds in *Thymus vulgaris* essential oil bound with all of these *F. oxysporum* proteins and conferred pathogenicity, whereas hexadecenoic acid, palmitic acid, tetratetracontane, and stigmast-5-en-3-ol (3á,24S) could bind with the FMK1, SET7, SEG1, and Rho1 proteins; these represent the compositions of the dichloromethane and chloroform extracts from *O. europaea* leaves. Hence, this leads to the inhibition of the pathogenicity of *F. oxysporum*. These proteins are vital elements of the pathway of a transduction signal that controls numerous *F. oxysporum* infection processes [33]. Several studies have explained the importance of specific pathogenicity proteins, e.g., FMK1, SET7, SET12, AreA, MeaB, Rho1, MKK1,2, SEG1, XINR, and Hog, in various pathways that are responsible for the virulence of *F. oxysporum* and that have a role in disease control [18,35]. These results are in agreement with those of [35], as they stated that molecular docking analyses were performed to clarify the antifungal effectiveness of the most and least active compounds against the Fgb1 and Fophy fungal proteins. This study indicates that dichloromethane, chloroform, and *O. europaea* leaves may be considered as the most important sources of antifungal compounds.

## 4. Materials and Methods

### 4.1. Morphological Characterization and Isolation of Fusarium oxysporum f. sp. lactucae

Fusarium isolate samples were collected from lettuce plants at the Faculty of Agriculture, Cairo University, Giza, Egypt. The stems of the infected samples were sterilized by filling them with 2% sodium hypochlorite solution for 4 min. They were then exhaustively washed with dH_2_O. Then, the samples were cut into 4 pieces of 5 mm of diseased tissue, which was transferred onto potato dextrose agar (PDA) media. The antibiotic streptomycin was added to the PDA media to decrease contamination resulting from bacterial growth. The fungal culture was incubated at 26 ± 2 °C and was periodically checked. The fungal growth was identified and purified for characterization and usage in the following experiments. A single spore of the fungal isolates was grown for 11 days on a PDA medium for morphological identification. The culture properties were detected from 11 to 15 days in the PDA cultures. The microscopic features of the chlamydospores and conidia were also detected by following the methods of previous reports [36,37].

### 4.2. Pathogenicity Test

A test of the pathogenicity of the *F. oxysporum* isolates was carried out at the greenhouse of the Plant Pathology Department, Faculty of Agriculture, Cairo University, Giza, Egypt. The pathogenicity of the fungal isolates was assessed in lactucae seedlings at the true leaf stage. The roots were filled with 60 mL of a suspension of conidia for three minutes. Then, the seedlings were transferred into sterilized soil in plastic pots and incubated under greenhouse conditions. A final determination of disease development was made after 20–30 days.

### 4.3. Molecular Identification of Fusarium oxysporum Isolates

#### 4.3.1. Genomic Extraction from Fungal DNA

The single, pure cultures of fungus were grown in a medium of potato dextrose broth (PDB) in darkness for 7 days at 25 °C. Mycelia were collected through purification with filter paper, then harvested in 0.85% NaCl saline solution. The collected mycelia were used directly for DNA isolation with the Gene JET Genomic DNA Purification Kit (Thermo Scientific, Lithuania, USA). The DNA yield and purity were checked using both a Nanodrop spectrophotometer and agarose gel electrophoresis.

#### 4.3.2. Identification of *F. oxysporum* Isolate through ITS Gene Sequencing

The internal transcribed spacer (ITS) region was identified with the universal primers of the ITS1 (5′-CTTGGTCATTTAGAGGAAGTAA-3′) and ITS4 (5′ TCCTCCGCTTATTGATATGC-3′) sequences. The amplification step was performed using a thermal cycler for PCR (Bio-Rad T100, Hercules, CA, USA). The PCR products were amplified through agarose gel electrophoresis and by using a gel extraction kit for purification; then, the purified PCR products were sent for sequencing by Macrogen (Seoul, Korea).

#### 4.3.3. Sequencing and Bioinformatic Analyses

The conserved sequence of the ITS gene of the studied fungus isolate’s genome was similar to the ITS sequences in the database, as established through the use of the Basic Local Alignment Search Tool (BLAST), which is found on the website of the NCBI (https://blast.ncbi.nlm.nih.gov, 2020). Then, this part of the *F. oxysporum* f. sp. *lactucae* sequence was compared with similar sequences of strains of *F. oxysporum* in the NCBI database; then, a phylogenetic tree was constructed by using the MEGA 6 software program (https://mega.software.informer.com/6.0/, 2020). The phylogenetic analysis was performed by using the maximum likelihood tree method. The tree distance was calculated using the maximum composite likelihood method.

### 4.4. Plant Extract Preparation

#### 4.4.1. Materials

The *T. vulgaris* and *O. europaea* leaves used in this work were obtained from the Medicinal, Aromatic, and Poisonous Plant Experimental Station, Faculty of Pharmacy, Cairo University, as well as the Department of Medicinal and Aromatic Plants, Faculty of Agriculture, Cairo University, Egypt.
All solvents used were of LC/MS grade and were purchased from J. T. Baker (The Netherlands).All other chemicals and standards were purchased from Sigma Aldrich (St. Louis, MO, USA).

#### 4.4.2. Preparation of *O. europaea* Leaf Extracts

The O. europaea leaves were dried at room temperature (26 ± 2 °C) for two weeks and ground to a fine powder. The ground leaves (1 g) were extracted with 10 mL of high-purity ethanol, dichloromethane, chloroform, and ethyl acetate solvents. Whatman filter paper no. 1 was used to filtrate the extracts, and then a rotary evaporator was used to concentrate the extracts at 40 °C by using 50 mL centrifuge tubes. The extracts were dried in a glassy desiccator; then, the residues of the powder were stored for analysis. Extraction was carried out three times with the same volume of solvent added repeatedly.

#### 4.4.3. Preparation of Essential Oil (*Thymus vulgaris*)

The fresh leaves were collected and subjected to hydrodistillation in a Clavenger apparatus for 5 h. according to the procedure described in the Egyptian Pharmacoepia [38].

### 4.5. Antifungal Activity of Plant Extracts and Essential Oils In Vitro

The antifungal activity of the ethanol, dichloromethane, chloroform, and ethyl acetate extracts from *O. europaea* leaves, as well as the oil of *T. vulgaris* and *B. carteri* were examined in vitro by using a poisoned food technique [39]. Each treatment was separately dissolved in DEMSO (1:1 *v/v*) to prepare stock of 75.5, 37.75 and 18.875 mg/mL for extracts and µL/mL for oils were added to 100 mL a sterile Erlenmeyer flask containing 60 mL cooled molten PDA, then rotated manually to disperse the solutions. We dispensed 20 milliliters of the medium into sterile Petri dishes (9 cm in diameter). The medium was allowed to solidify at room temperature. Agar discs taken from the margins of fresh and pure culture were aseptically inoculated at the center of the Petri plates. Control plates (with DMSO only) were inoculated following the same procedure. The fungicide nystatin (100,000 units/mL) was used as a positive control. We tested with 200 microliters in PDA medium. The nystatin was purchased from Sigma Aldrich (St. Louis, MO, USA). The tests were performed in triplicate, and all plates were incubated at 25 °C. We measured the fuserium radial growth after 6 days of incubation at 25 °C. All experiments were performed in triplicate for each treatment. The average of the two orthogonal diameters was measured when fungal mycelium covered one plate in control treatment.

### 4.6. GC/MS Analysis of Essential Oil (*Thymus vulgaris*)

The mass spectra were recorded with a Shimadzu GCMS-QP2010 (Tokyo, Japan) equipped with a split–splitless injector and Rtx-5MS column (30 m × 0.25 mm i.d. × 0.25 µm film thickness) (Restek, Bellefonte, PA, USA). The capillary column was attached to a quadrupole mass spectrometer (SSQ 7000; Thermo-Finnigan, Bremen, Germany). The initial temperature of the column was set to 45 °C for 2 min and programmed to 300 °C at a rate of 5 °C/min; then, it was kept constant at 300 °C for 5 min. The injector temperature was 250 °C. The flow rate of the carrier gas (helium) was 1.41 mL/min. The mass spectra were recorded according to the following conditions: filament emission current (equipment current), 60 mA; ionization voltage, 70 eV; ion source, 200 °C. Diluted samples (1% *v/v*) were injected via the split mode (split ratio: 1:15) [33].

#### Sample Sialylation for GC/MS Analysis

For the analysis of primary metabolites in different samples, a derivatization step was performed before the analysis, as described in Farag et al. (2015) [40]. Briefly, the extract (50 μL) was mixed with 100 μL of N-methyl-N-(trimethylsilyl)-trifluoroacetamide (MSTFA) and incubated at 60 °C for 45 min. Samples were equilibrated at 28 °C and analyzed using a Shimadzu model QP-5000 GC-MS mass spectrometer (Kyoto, Japan).

The silylated derivatives were separated on an Rtx-5MS column, and all injections were performed in the 1:15 split mode. The quadrupole mass spectrometer was operated in an electron ionization mode at 70 eV. The scan range was set to 50–650 *m/z*.

### 4.7. Molecular Docking Analysis

In this investigation, the molecular docking of the tested phytochemical compounds extracted from *T. vulgaris* and *O. europaea* leaves with eleven essential proteins involved in *F. oxsyporum* virulence, i.e., AreA, MeaB, Fmk1, Ste7, Set12, Sge1, Xin R, Hog1, PacC, Mkk12, and Rho1, was studied. The molecular docking analysis was performed using the SAMSON 2020 software (https://www.samson-connect.net/) to determine the interactions between the target virulence proteins of Fusarium and the ligand structures of the tested plant extract compounds and to identify the direct effects of these compounds on the inhibition of Fusarium. The sequence of each protein was downloaded from NCBI (https://www.ncbi.nlm.nih.gov/) in the FASTA format to build binding models with a 3D structure by using the TASSER server (https://zhanglab.ccmb.med.umich.edu/I-TASSER/). The SWISSMODLE server (https://swissmodel.expasy.org/) was used for the construction of the 3D proteins. The affinity minimization was performed using the 3DREFINE server (http://sysbio.rnet.missouri.edu/3Drefine/index.html). In pre-docking, all water molecules and ligands were deleted, and the hydrogen atoms were added to the target proteins. On the other hand, the ligands were downloaded from PubChem (https://pubchem.ncbi.nlm.nih.gov/) in the SDF format, and then the openbabel software (http://openbabel.org/wiki/Main_Page) was used for the conversion into the MOL2 format. The interactions of the Fusarium proteins were built into the models with the ligand structures of the ethanol, dichloromethane, chloroform, and ethyl acetate extracts of *O. europaea* leaves, as well as the oil of *T. vulgaris*. The docking of the proteins with the tested compounds was performed with the aid of the SAMSON 2020 software. The calculations of the free binding energies were performed by using the scoring function of AutoDock Vina as an element in its script. Following an exhaustive search, 100 poses were examined, and the best-scoring poses were selected to compute the binding energies of the ligands. In addition, Discovery Studio (https://www.discngine.com/discovery-studio) was used for the 2D structures of the ligands.

### 4.8. Statistical Data Interpretation

Data analysis and graphs were made using the GraphPad Prism Version 9 program. The data are expressed as an arithmetic mean, standard deviation and 95 percent confidence interval for the IC50 parameter. IC50 was calculated as a concentration of the tested compound which decreases the mycelial growth by half between the base and maximum. *p* values less than or equal 0.05 were considered statistically significant [41].

## 5. Conclusions

The molecular identification of *F. oxysporium*, which causes Fusarium wilt in lactucae plants in Egypt, was achieved. In this investigation, the dichloromethane chloroform and ethanol extracts of *O. europaea* leaves and the *T. vulgaris* essential oil showed strong effects against *F. oxysporum*. The antifungal screening against *F. oxysporum* f. sp. *lactucae* (*F.o.L*) verified that the main active ingredients of these extracts displayed considerable antifungal activity. In this respect, the results of the molecular docking analysis showed that the carvacrol, α-thujene, and thymol compounds in *Thymus vulgaris* essential oil bound specifically with eleven pathogenic proteins in *F. oxysporium*, whereas hexadecenoic acid, palmitic acid, tetratetracontane, and stigmast-5-en-3-ol (3á,24S) bound with the FMK1, SET7, SEG1, and Rho1 proteins. In particular, the chloroform extract of *O. europaea* leaves was selected as a source of antifungal substances for use against *F. oxysporum,* and it could provide a new lead in the pursuit of new biological sources of agrochemical candidates.

## Figures and Tables

**Figure 1 molecules-26-06118-f001:**
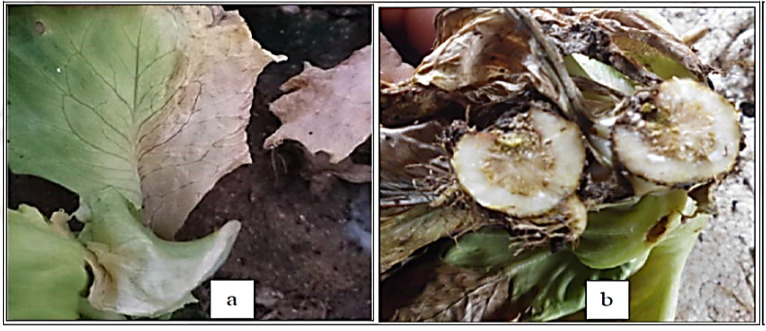
Symptoms of Fusarium wilting in lettuce under artificial infection symptoms on the margin of the outer leaves (**a**) and brown discoloration in the crown area (**b**).

**Figure 2 molecules-26-06118-f002:**
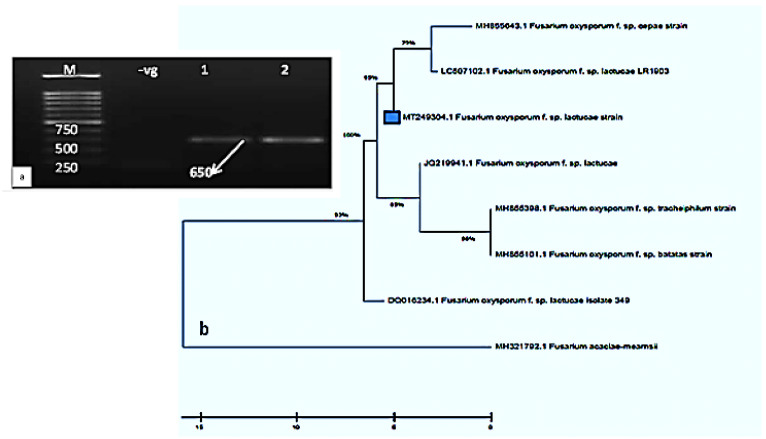
PCR product and phylogenetic tree of the *Fusarium oxysporum* f. sp. *lactucae* isolate. (**a**) PCR product of the isolate. M = Molecular marker, –vg = Negative control, (1–2) = *F. oxysporum* samples. (**b**) Phylogenetic tree based on the ITS gene sequences of the isolates.

**Figure 3 molecules-26-06118-f003:**
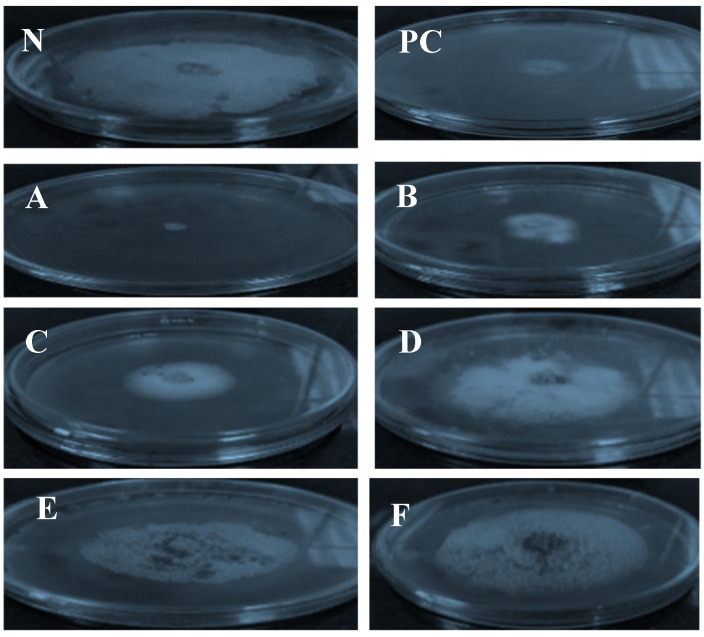
Effects of *T. vulgaris* oil (A), chloroform (B), dichloromethane (C), ethanol (D) and ethyl acetate extracts (E) of *O. europaea* leaves and B. carteri (F) on mycelium linear growth of. *F. oxysporum* f. sp. *lactucae* after six days of cultivating the fungus on PDA medium at 25 °C. NC: negative control (DMSO), PC: positive control (Nystatin).

**Figure 4 molecules-26-06118-f004:**
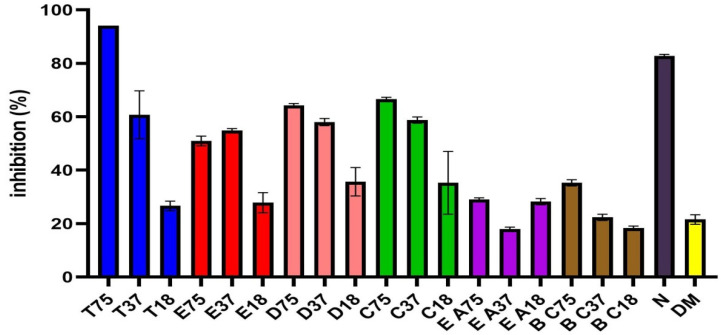
Mycelial growth % inhibition with different concentrations of T: thyme oil, E: ethanol, D: dichloromethane, C: chloroform, EA: ethyl acetate, B C: *B. carteri*, N: nystatin, DM: DMSO.

**Figure 5 molecules-26-06118-f005:**
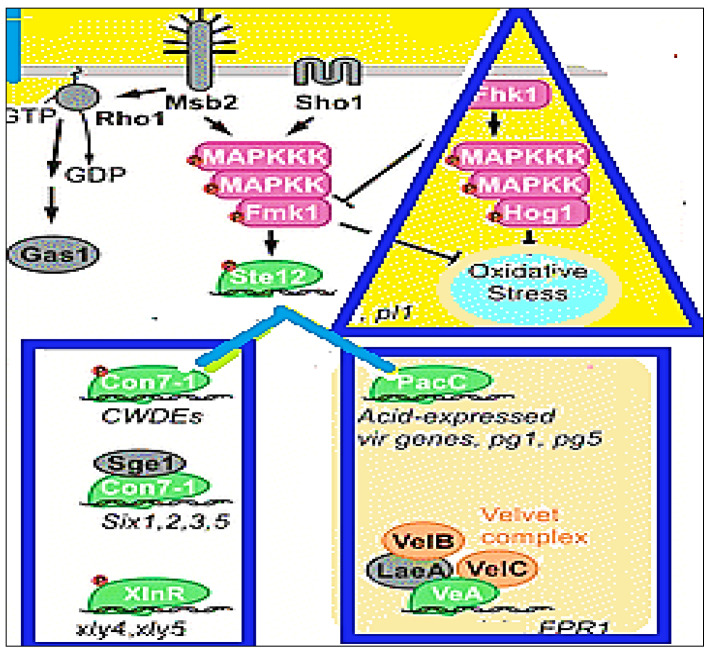
The pathway of the major protein kinases and transcription factors in the infection process of *F. oxysporum* [18].

**Figure 6 molecules-26-06118-f006:**
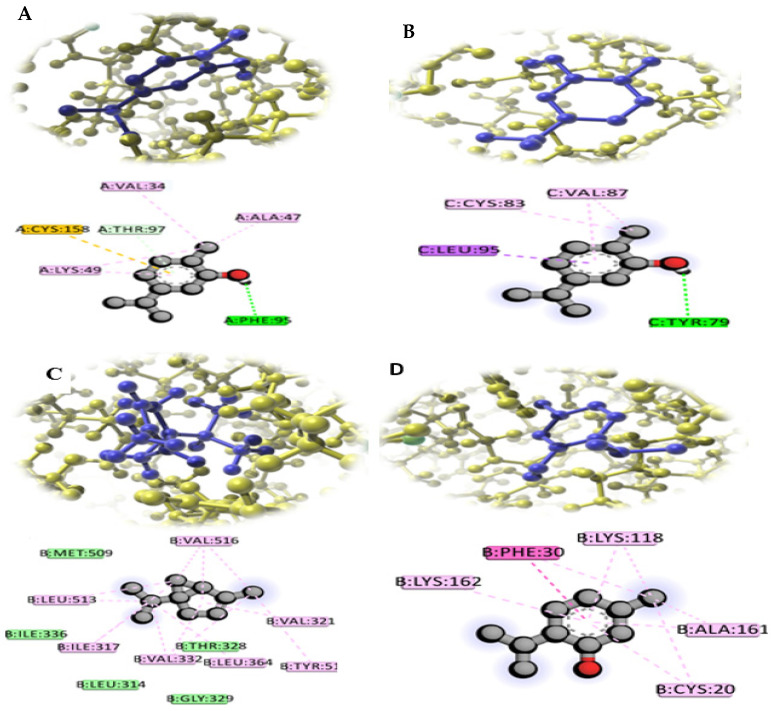
The 2D and 3D interaction diagrams of the binding of carvacrol with Hog1 (**A**) and PacC (**B**), the binding of α-thujene with Mkk12 (**C**) and Thymol, and the binding of Rho1 (**D**) with Fusarium virulence proteins.

**Table 1 molecules-26-06118-t001:** IC_50_ of different plant extracts against *F. oxysporum*.

Plant Extract	IC_50_ (mg/mL)
Thyme oil	28.99 (µL/ml)
Ethanol extract	49.60
Dichloromethane extract	32.51
Chloroform extract	31.35
Ethyl acetate extract	521.9
B. carteri	176.2

**Table 2 molecules-26-06118-t002:** Composition of the dichloromethane extract from the *O. europaea* leaves according to the GC-MS analysis.

Compound No.	RI	Identified Compound	MWT	%Similarity	Chemical Class	Area%
**1**	788	1,2-Bis(trimethylsiloxy)ethane	206	75	Alkane	1.3
**2**	905	Ethanimidic acid, N-(trimethylsilyl)-, trimethylsilyl ester	203	93	Acid	2.2
**3**	915	Propanoic acid, 2-[(trimethylsilyl)oxy]-, trimethylsilyl ester	234	96	Acid	1.8
**4**	1078	Butanoic acid, 4-[(trimethylsilyl)oxy]-, trimethylsilyl ester	248	89	Acid	1.6
**5**	1086	Cyclohexanone, 3,3,5-trimethyl-	140	77	Ketone	0.4
**6**	1108	Glycerol, 3TMS derivative	308	96	Alcohol	6.9
**7**	1163	Picolinic acid, TMS derivative	195	95	Acid	2.2
**8**	1192	Octanoic acid, trimethylsilyl ester	216	75	Fatty acid	0.2
**9**	1252	2-(2-Butoxyethoxy)ethoxy-trimethylsilane	254	98	Alcohol	0.9
**10**	1338	3,6,9,12-Tetraoxa-2,13-disilatetradecane, 2,2,13,13-tetramethyl-	294	70	Alcohol	0.4
**11**	1406	Octanoic acid, TBDMS	258	76	Fatty acid	0.2
**12**	1612	Hexadecane	226	95	Alkane	0.4
**13**	1664	8,10-Dioxaheptadecane	244	87	Alkane	3.8
**14**	1685	L-Proline, 2TBDMS derivative	343	73	Amino acid	0.9
**15**	1699	L-Leucine, 2TBDMS derivative	359	87	Amino acid	3.6
**16**	1699	Isoleucine, 2TBDMS derivative	359	89	Amino acid	1.3
**17**	1769	Myristic acid	228	93	Fatty acid	0.4
	1788	Tetradecanoic acid, trimethylsilyl ester	300	85	Fatty acid	0.4
**19**	1878	Hexadecenoic acid, methyl ester	270	87	Fatty acid	0.7
**20**	1916	Mandelic acid di(tert-butyldimethylsilyl)-	380	91	Acid	3.4
**21**	1968	Palmitic acid	256	92	Fatty acid	11.6
**22**	1987	Hexadecenoic acid, trimethylsilyl ester	328	77	Fatty acid	5.8
**23**	2009	Eicosane	282	97	Alkane	0.4
**24**	2037	α.-D-(+)-Talopyranose, pentakis(trimethylsilyl) ether	540	85	Sugar	0.4
**25**	2066	Trimethylsilyl ether of glucitol	614	70	Sugar	1.6
**26**	2081	7,9-Di-tert-butyl-1-oxaspiro(4,5)deca-6,9-diene-2,8-dione	276	88	Lactone	0.9
**27**	2186	Octadecanoic acid, trimethylsilyl ester	356	85	Fatty acid	3.8
**28**	2194	Oleic acid, trimethylsilyl ester	354	86	Fatty acid	2.5
**29**	2581	Hexadecanoic acid, 2,3-bis[(trimethylsilyl)oxy]propyl ester	474	94	Fatty acid	1.3
**30**	2780	Glycerol monostearate, 2TMS derivative	502	85	Alcohol	0.4
**31**	2789	β.-Sitosterol trimethylsilyl ether	486	87	Sterol	1.1
**32**	2900	Nonacosane	408	95	Alkane	0.4
**33**	2914	Squalene	410	86	Organic compound	0.4
**34**	2931	α.-Amyrin, TMS derivative	498	96	Triterpene	0.9
**35**	3343	Spirosol-5-en-3-ol, 28-acetyl-, acetate, (3.β.,22.α.,25R)-	497	97	Alcohol	7.2
**36**	3401	Tetratriacontane	478	95	Alkane	1.6
**37**	3410	Stigmasta-3,5-dien-7-one	410	85	Sterol	0.7
**38**	3600	Hexatriacontane	506	95	Alkane	3.8
**39**	3986	Propanoic acid, 3,3′-thiobis-, ditetradecyl ester	570	70	Acid	2.7

**Table 3 molecules-26-06118-t003:** Composition of the chloroform extract from the *O. europaea* leaves according to the GC-MS analysis.

Compound No.	RI	Identified Compound	Chemical Class	MWT	%Similarity	Area%
**1**	788	1,2-Bis(trimethylsiloxy)ethane	Alkane	206	84	0.9
**2**	905	Ethanimidic acid, N-(trimethylsilyl)-, trimethylsilyl ester	Acid	203	95	1.2
**3**	915	Propanoic acid, 2-[(trimethylsilyl)oxy]-, trimethylsilyl ester	Acid	234	95	0.5
**4**	993	Hexanoic acid, TMS derivative	Acid	188	78	0.7
**5**	1078	Butanoic acid, 4-[(trimethylsilyl)oxy]-, trimethylsilyl ester	acid	248	89	1.4
**6**	1086	Cyclohexanone, 3,3,5-trimethyl-	Ketone	140	75	0.2
**7**	1108	Glycerol, 3TMS derivative	Alcohol	308	95	7.3
**8**	1163	Picolinic acid, TMS derivative	Acid	195	96	2.1
**9**	1192	Octanoic acid, trimethylsilyl ester	Fatty acid	216	73	0.2
**10**	1252	2-(2-Butoxyethoxy)ethoxy-trimethylsilane	Alcohol	254	97	0.9
**11**	1338	3,6,9,12-Tetraoxa-2,13-disilatetradecane, 2,2,13,13-tetramethyl-	Alcohol	294	70	0.2
**12**	1406	Octanoic acid, TBDMS	Fatty acid	258	72	0.2
**13**	1437	3,6,10,13-Tetraoxa-2,14-disilapentadecane, 2,2,14,14-tetramethyl-	Alcohol	308	70	0.2
**14**	1585	Quinoline	Alkaloid	197	73	0.7
**15**	1612	Hexadecane	alkane	226	70	0.5
**16**	1664	8,10-Dioxaheptadecane	Alkane	244	89	3.1
**17**	1682	Hexadecane, 7,9-dimethyl-	Alkane	254	90	0.2
**18**	1685	L-Proline, 2TBDMS derivative	Amino acid	343	71	0.7
**19**	1692	β.-D-(+)-Xylopyranose, 4TMS derivative	Sugar	438	86	0.2
**20**	1699	L-Leucine, 2TBDMS derivative	Amino acid	359	90	0.9
**21**	1699	Isoleucine, 2TBDMS derivative	Amino acid	359	88	0.2
**22**	1766	Sebacic acid, 2TMS derivative	Acid	346	80	0.9
**23**	1769	Myristic acid	Fatty acid	228	92	0.2
**24**	1788	Tetradecanoic acid, trimethylsilyl ester	Fatty acid	300	76	0.9
**25**	1878	Hexadecanoic acid, methyl ester	Fatty acid	270	94	0.5
**26**	1888	n-Pentadecanoic acid, trimethylsilyl ester	Fatty acid	314	85	0.5
**27**	1916	Mandelic acid di(tert-butyldimethylsilyl)-	Acid	380	94	2.8
**28**	1968	Palmitic acid	Fatty acid	256	92	9.2
**29**	1987	Hexadecanoic acid, trimethylsilyl ester	Fatty acid	328	80	8.5
**30**	2009	Eicosane	Alkane	282	96	0.2
**31**	2037	α.-D-(+)-Talopyranose, pentakis(trimethylsilyl) ether	Sugar	540	87	0.7
**32**	2066	Trimethylsilyl ether of glucitol	Sugar	614	92	1.9
**33**	2067	Heptadecanoic acid	Fatty acid	270	75	0.9
**34**	2081	7,9-Di-tert-butyl-1-oxaspiro(4,5)deca-6,9-diene-2,8-dione	Lactone	276	90	0.5
**35**	2087	Heptadecanoic acid, TMS derivative	Fatty acid	342	85	0.5
**36**	2186	Octadecanoic acid, trimethylsilyl ester	Fatty acid	356	75	5.0
**37**	2194	Oleic acid, trimethylsilyl ester	Fatty acid	354	75	5.2
**38**	2210	α.-Linolenic acid, TMS derivative	Fatty acid	350	90	0.2
**39**	2581	Hexadecanoic acid, 2,3-bis[(trimethylsilyl)oxy]propyl ester	Fatty acid	474	85	2.8
**40**	2780	Glycerol monostearate, 2TMS derivative	Alcohol	502	87	0.5
**41**	2788	1-Monooleoylglycerol, 2TMS derivative	Alcohol	500	70	0.2
**42**	2789	β.-Sitosterol trimethylsilyl ether	Sterol	486	71	1.2
**43**	2796	1-Monolinolein, 2TMS derivative	Fatty acid	498	91	14.2
**44**	2900	Nonacosane	Alkane	408	89	0.2
**45**	2914	Squalene	Organic compound	410	95	0.5
**46**	2931	α.-Amyrin, TMS derivative	Triterpene	498	96	1.2
**47**	2955	Ergometrinine, 2TMS derivative	Alkaloid	469	70	0.5
**48**	3049	Tris(4-bromophenyl) amine	Amine	479	86	0.2
**49**	3228	Catechin (2R-E)-, 5TMS derivative	Flavonoid	650	72	0.9
**50**	3343	Spirosol-5-en-3-ol, 28-acetyl-, acetate, (3.β.,22.α.,25R)-	Alcohol	497	96	6.6
**51**	3401	Tetratriacontane	Alkane	478	96	0.5
**52**	3410	Stigmasta-3,5-dien-7-one	Sterol	410	75	0.7
**53**	3600	Hexatriacontane	Alkane	506	95	2.8
**54**	3986	Propanoic acid, 3,3′-thiobis-, ditetradecyl ester	Acid	570	75	2.8

**Table 4 molecules-26-06118-t004:** Composition of the *T. Vulgaris* essential oil extract according to the GC-MS analysis.

Compound No.	RI	Identified Compounds	%Similarity	Area (%)
**1**	930	α-Thujene	97	1.4
**2**	948	α-Pinene	97	0.8
**3**	952	Camphene	97	0.4
**4**	975	Sabinen	97	0.1
**5**	979	β-Pinene	98	0.3
**6**	991	β-Myrcene	96	1.6
**7**	1009	α-Phellandrene	94	0.3
**8**	1013	δ-3-Carene	95	0.1
**9**	1018	α-terpinene	97	1.5
**10**	1020	o-Cymene	97	11.8
**11**	1059	p-Cineole	96	1.8
**12**	1062	γ-Terpinene	97	10.9
**13**	1109	2-p-Menthen-1-ol	96	1.0
**14**	1112	Linalool	97	1.3
**15**	1138	Borneol	97	1.0
**16**	1177	4-Terpineol	96	0.4
**17**	1231	p-Cymene	96	7.6
**18**	1262	Thymol	97	41.9
**19**	1298	carvacrol	96	3.6
**20**	1464	Caryophyllene	98	1.2
**21**	1515	Germacrene D	93	0.7
**22**	1524	δ-Cadinene	92	0.1
**23**	1581	Caryophyllene oxide	94	0.3
**24**	1757	Humulane-1,6-dien-3-ol	93	0.2

**Table 5 molecules-26-06118-t005:** Effects of plant phytochemical compounds, essential oils, and extracts on the *F. oxysporum* virulence proteins according to the docking analysis.

Protein/X Y Z	Ligand Name	Pubchem ID	Types of Bond	Score
FMK1 x = 27.4555 y = 1.5129 z = 40.778	Carvacrol	10364	H bond pi-sigma pi-alyl	−6.4
	Thymol	6989		−6.3
	α-Thujene	17868		−5.7
	Hexatriacontane	12412		−6
	Nonacosane	12409		−5.9
	Palmitic acid (Hexadecenoic acid)	985		−5.1
	Tetratriacontane	5282743		−5.7
	Stigmasta-3,5-dien-7-one	26519		−5.8
SET7 x = 47.3052 y = −15.6381 z = −5.1291	α-Thujene	17868	Hbond carbon Hbond pi-sulfur pi-Alkyl	−5.6
	Carvacrol	10364		−6.1
	Thymol	6989		−6.9
	Hexatriacontane	12412		−6.5
	Nonacosane	12409		−5.9
	Palmitic acid	985		−5
	(Hexadecenoic acid)	5282743		−5.2
	Tetratriacontane	26519		−5.8
	Stigmasta-3,5-dien-7-one	12444466		−4.2
SET12 x = 37.1282 y = 38.5316 z = 37.5226	α-Thujene	17868	H bond carbonHbond pi-sulfur pi-donor Hydrogen	−3.6
	Carvacrol	10364		−4.1
	Thymol	6989		−4.1
AreA x = 37.1282 y = 38.5316 z = 37.5226	α-Thujene	17868	H bond carbonHbond pi-sulfur pi-donor Hydrogen	−3.6
	Carvacrol	10364		−4.1
	Thymol	6989		−4.1
AreA x = 36.9175 y = −94.8871 z = 8.6668	α-Thujene	17868	Hbond Pi-pi-T-shaped	4.3
	Carvacrol	10364		−4.8
	Thymol	6989		−4.6
MeaB x = 16.164 y = 0.1875 z = −2.9961	Carvacrol	10364	carbon H pi-sigma pi-sulfur	−3.9
	Thymol	6989		−4.3
	α-Thujene	17868		−4.1
Rho1 x =94.6544 y =35.8305 z = 26.0388	α-Thujene	17868	Pi-pi-T-shaped pi-Alkyl	−5.2
	Carvacrol	10364		−5.5
	Thymol	6989		−6.5
	Hexatriacontane	12412		−5.8
	Nonacosane	12409		−5.4
	Palmitic acid	985		−5.2
	(Hexadecenoic acid)	5282743		−5.2
	Tetratriacontane	26519		−5.7
	Stigmasta-3,5-dien-7-one	12444466		−5.1
MKK1,2 x = 61.2779 y = −19.0646 z = 15.7687	α-Thujene	17868	Van der Waals Alkyl pi-Alkyl	−6.8
	Carvacrol	10364		−6.1
	Thymol	6989		−5.8
SEG1 x = 6.2314 y = 35.6614 z =101.8374	α-Thujene	17868	Conventional Hbond pi-sulfur pi-pi stacked	−4.1
	Carvacrol	10364		−4.9
	Thymol	6989		−5.2
	Hexatriacontane	12412		−5.1
	Nonacosane	12409		−5.4
	Palmitic acid	985		−5.1
	(Hexadecenoic acid)	5282743		−4.9
	Tetratriacontane	26519		−4.8
	Stigmasta-3,5-dien-7-one	12444466		−4.7
XINR x = −0.4654 y = −1.4131 z = −3.89	Thymol	6989	pi-donor H bond pi-Alkyl carbon H bond	−4.2
	α-Thujene	17868		−4.1
PACC x = 5.5317 y = 8.243 z = 49.2927	α-Thujene	17868	H bond pi-Alkyl pi-sigma	−4.2
	Carvacrol	10364		−5.2
	Thymol	6989		−4.5
Hog1 x = 26.713 y = 0.6519 z = 29.6776	Carvacrol	10364	H bond pi-sigma pi-sulfur pi-Alkyl, Alkyl	−6.3
	Thymol	6989		−5.7
	α-Thujene	17868		−5.8

## Data Availability

*Fusarium oxysporum* f. sp. *lactucae* from plant pathology department, Faculty of Agriculture, Cairo University, Giza, Egypt.

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
