# Peer review of "Antifungal Evaluation and Molecular Docking Studies of Olea europaea Leaf Extract, Thymus vulgaris and Boswellia carteri Essential Oil as Prospective Fungal Inhibitor Candidates"

_molecules, 2021, doi:10.3390/molecules26206118_

Round 1

Reviewer 1 Report

Major

The article is poorly written and need extensive editing. The compounds identified are questionable and cannot rely solely on the name given by the GC libraries. Some of these compounds should be carefully re-evaluated. Some journal titles in the reference are written in full while other are abbreviated. The retention time should not be included in the Table, only the RRI. Only one decimal should be used when reporting on the relative percentage area. Some values should be added in the abstract. The zone of inhibition is no longer used to evaluate the antibacterial activity. On what basis the samples selected? A clear connection must be provided. The reference is not cited correctly and the authors should consult the “guide lines of the journal”. No positive control is included in Table 1, this is the most basic element for any biological activity. It is indicated in the Table that the values and the mean +/- SD, but no standard deviation is provided. In addition, the authors should indicate how many replicates were conducted in the Table ( n = ???). I still have SERIOUS reservation on the identification of the compounds using GC-MS. The identity of the compounds provided are those given by the libraries which in most of cases here are in correct. The authors should use symbol (α, β ) and not word when reporting the composition of the extract/oil. Siloxane derivative may be the result of column bleeding and not compounds found in the extract. The RI calculated and previous RI should be reported for better comparison. No retention time should be provide when reporting GC MS data

L26: “the active ingredient” how did the authors came to find out that these were the active ingredients in the plant. This conclusion is just a pure speculation

L61L T. vulgaris , thymus should be written in full at the beginning of a sentence

L64-65: O. 64 europaeaI Olea should be written in full at the beginning of a sentence

L137, 138: “….vulgaris…’should be with ‘v” and not “V”

143: T. vulgaris should be in italics

Table 1, should be DMSO and not DEMSO

Author Response

Authors reply list

Manuscript Title: Antifungal Evaluation and Molecular Docking Studies of Olea europaea leaves extract and Thymus vulgaris essential oil as Prospective Fungal Inhibitor Candidates

Corresponding Author:

Professor. Soheir N. Abd El-Rahman , Dr. Hanaa S. Omar

Email: Soheirkenawy@yahoo.com, Hanaa8324@yahoo.com

Dear reviewer,

We are grateful to the reviewer for their time and constructive comments on our manuscript. We think reviewers’ constructive comments greatly improved manuscript. We have carefully addressed all the comments. The corresponding changes and refinements made in the revised paper are summarized in our response below.

Comments from reviewers:
-Reviewer1
Dear reviewer1, allow me to appreciate your very valuable and precious comments which considerably helped to improve and strengthen the manuscript. We are honored to offer you the following replies hoping that you will find them satisfactory and properly addressing your concerns. We have addressed the reviewer’s suggestions and revised the manuscript accordingly. Please find attached a detailed point-by-point response to reviewer's concerns.

  1. article is poorly written and need extensive editing

Response: We agree with the reviewer.

We revised the article and corrected it using grammar checker (Grammarly premium). We wonder if we can use your English editing service to improve our manuscript.

  1. The compounds identified are questionable and cannot rely solely on the name given by the GC libraries.

Response: We thank the reviewer for the kind comment.

After the rejection and resubmission status of our manuscript, we performed sialylation of plant samples. We reanalyzed silylated plant extracts using GCMS at the faculty of pharmacy Ain Shams university. Identification of compounds done by comparing mass spectrum and retention index of compounds using NIST library with the aid of review data collected about plants under study. 

  1. Some of these compounds should be carefully re-evaluated.

Response: We thank the reviewer for the valuable comment.

We revised the identified compounds carefully by comparing mass spectrum and retention index.

  1. Some journal titles in the reference are written in full while other are abbreviated

Response: We agree with the reviewer

We revised all references accurately and corrected it.

  1. The retention time should not be included in the Table, only the RRI.

Response: We thank the reviewer for the constructive comment.

We modified it as required. The retention time was removed.

  1. Only one decimal should be used when reporting on the relative percentage area.

Response: We thank the reviewer for the kind comment.

We edited the relative percentage area as required.

7. Some values should be added in the abstract.

Response: We thank the reviewer for the valuable comment.

We modified the abstract in the manuscript as follow

dichloromethane efficiently inhibited the growth of the test, dichloromethane extract exhibited the maximum zone of inhibition (34.83 mm). Whereas, ethanol, ethyl acetate and B. carteri EO didn’t (line 22&23)

and palmitic acid (41.9, 3.6, 0.7, 5.2), respectively. Moreover, molecular modeling revealed    (line 26)

  1. The zone of inhibition is no longer used to evaluate the antibacterial activity.

Response: We thank reviewer for the valuable comment.

The zone of inhibition was determined according to some references as they used this method in evaluation of antibacterial and antifungal activity. (1, 2 and 3). Moreover, there is variation between extract treatments compared with positive and negative control according to the well diffusion method. 

  1. On what basis the samples selected? A clear connection must be provided.

Response: We thank reviewer for the valuable comment.

 Natural extracts that have not been used against F. oxysporum forma sp. Lettuce (F.O.l) 

  1. The reference is not cited correctly and the authors should consult the “guide lines of the journal”.

Response: We thank reviewer for the valuable comment.

 In response to this comment, we edited it the correct way in all references section.

11.No positive control is included in Table 1, this is the most basic element for any biological activity.

Response: we agree with the reviewer. So, the positive and negative controls were added in table1 and result section as suggested by the reviewer.

  1. It is indicated in the Table that the values and the mean +/- SD, but no standard deviation is provided.

Response: We thank the reviewer. We edited it.

  1. In addition, the authors should indicate how many replicates were conducted in the Table ( n = ???).

Response: We thank the reviewer. We edited it.

  1. I still have SERIOUS reservation on the identification of the compounds using GC-MS. The identity of the compounds provided are those given by the libraries which in most of cases here are in correct.

Response: We thank the reviewer for the valuable and constructive comment.

After our manuscript's rejection and resubmission status, samples were completely reanalyzed. we performed sialylation of plant samples using N-methyl-N-(trimethylsilyl)-trifluoroacetamide (MSTFA) as mentioned in the material and methods section. We reanalyzed silylated plant extracts using GCMS at the faculty of pharmacy Ain Shams university. Identification of compounds done by comparing mass spectrum and retention index of compounds using NIST library with the aid of review data collected about plants under study. We didn't put compounds given by the library as it is.  We carefully compared the retention index and mass spectrum of compounds with reference data to confirm identification.

15. The authors should use symbols (α, β ) and not word when reporting the composition of the extract/oil.

Response: We thank the reviewer for the kind comment

We modified it as required by the reviewer

  1. Siloxane derivative may be the result of column bleeding and not compounds found in the extract.

Response: We agree with the reviewer

We removed all siloxane derivatives as suggested by the reviewer

  1. The RI calculated and previous RI should be reported for better comparison.

Response: We thank the reviewer for the kind comment

We accurately compared RI of all compounds with their reference RI and confirmed that all numbers correct and within the range

  1. No retention time should be provide when reporting GC MS data

Response: We thank the reviewer for the valuable comment

We removed retention time as suggested by the reviewer

  1. L26: “the active ingredient” how did the authors came to find out that these were the active ingredients in the plant.

Response: We thank the reviewer for the valuable comment

We found that they are potent antifungal compounds from the previous research studies as Principal components, thymol, and carvacrol play a vital role in the antifungal activity of the oil 4. Molecular modeling confirmed that these compounds showed potent activity with higher scores than other compounds. docking studies revealed that there are some active ingredients in these extracts have a role in prevention the Fusarium infection process.

  1. L61L T. vulgaris , thymus should be written in full at the beginning of a sentence L64-65: O. 64 europaeaI Olea should be written in full at the beginning of a sentence

Response: We thank the reviewer. We edited it.

  1. L137, 138: “….vulgaris…’ should be with ‘v” and not “V” 143: T. vulgaris should be in italics.

Response: We thank the reviewer. We edited it.

  1. Table 1, should be DMSO and not DEMSO.

Response: We thank the reviewer. We edited it.

Reviewer 2 Report

The work presented in this manuscript is interesting and merits publication after addressing the following comments. 
1.        The abbreviation DEMSO presented in Table 1 is not explained anywhere in the manuscript.
2.        Figure 3: The letter G is not presented anywhere in the figure despite it exists at the legend.
3.        There is a question regaring the antifungal properties. The authors show some interesting results of the dichloromethane, ethanol, chloroform and ethyl acetate extracts. Though they haven’t taken into consideration the possibility that pure solvents may cause an inhibition to the fungus' growth. I would recommend the authors to run an experiment under the same conditions with pure solvents in the same amount used with essential oils to find out the inhibition caused by them and add these data in Table 1. This is mostly supported by the fact that the authors state in lines 234-235 that ethanol and ethyl acetate extracts didn’t exhibit antifungal effect instead of chloroform and dichloromethane. These last two solvents are chlorinated solvents and as a result they are usually more toxic to cells.  

Author Response

Manuscript Title: Antifungal Evaluation and Molecular Docking Studies of Olea europaea leaves extract and Thymus vulgaris essential oil as Prospective Fungal Inhibitor Candidates

Corresponding Author:

Professor. Soheir N. Abd El-Rahman , Dr. Hanaa S. Omar

Email: Soheirkenawy@yahoo.com, Hanaa8324@yahoo.com

-Reviewer2
Dear reviewer2, allow me to appreciate your very valuable and precious comments which considerably helped to improve the manuscript. We are honored to offer you the following replies hoping that you will find them satisfactory and properly addressing your concerns. We have addressed the reviewer’s suggestions and revised the manuscript accordingly. Please find attached a detailed point-by-point response to reviewer's concerns.

  1. The abbreviation DEMSO presented in Table 1 is not explained anywhere in the manuscript.

Response: We thank the reviewer for the kind comment.

In response to this comment we edited it the correct way in table1.  

  1. Figure 3: The letter G is not presented anywhere in the figure despite it exists at the legend.

Response: We thank the reviewer. We edited it.

  1. There is a question regarding the antifungal properties. The authors show some interesting results of the dichloromethane, ethanol, chloroform, and ethyl acetate extracts. Though they haven’t taken into consideration the possibility that pure solvents may cause inhibition to the fungus' growth. I would recommend the authors to run an experiment under the same conditions with pure solvents in the same amount used with essential oils to find out the inhibition caused by them and add these data in Table 1. This is mostly supported by the fact that the authors state in lines 234-235 that ethanol and ethyl acetate extracts didn’t exhibit antifungal effects instead of chloroform and dichloromethane. These last two solvents are chlorinated solvents and as a result, they are usually more toxic to cells.

Response: We thank the reviewer for the valuable and constructive comment.

Plant extracts with different solvents were evaporated under reduced pressure at a temperature not exceeding 40°C to obtain dried extracts residues free from solvents. Dried extracts residues were dissolved in DMSO so we tested DMSO as a negative control.

Reviewer 3 Report

The manuscript still requires some corrections:

  • Table 1 notes "values are means of three replicates +- SD", but I can't see SD in the table.
  • Table 2-4. How many repetitions were performed and what was SD for GC area?
  • Table 2-4. Calculations of RI in some cases ae not correct. Two compounds with different retention time, cannot have the same RI, for example, RI 1252 both for 10.452 min and 11.356 min (Table 2). The same can be find in other tables.
  • Which method was used for T.vulgaris essential oil: described in 4.5.1 or in 4.5.3? The sample preparation procedure still is not clear: what amount of plant material and solvent was used?
  • Producers and purity of all chemical compounds should be mentioned in the paper.

Author Response

- Authors reply list

Manuscript Title: Antifungal Evaluation and Molecular Docking Studies of Olea europaea leaves extract and Thymus vulgaris essential oil as Prospective Fungal Inhibitor Candidates

Corresponding Author:

Professor. Soheir N. Abd El-Rahman , Dr. Hanaa S. Omar

Email: Soheirkenawy@yahoo.com, Hanaa8324@yahoo.com

Reviewer 3

Dear reviewer, we thank the reviewer for their careful reading of the manuscript and their constructive remarks. We have taken the comments on board to improve and clarify the manuscript. Please find below a detailed point-by-point response to all comments

  1. values are means of three replicates +- SD", but I can't see SD in the table

Response: We thank the reviewer for the valuable comment.

In response to this comment we edited it the correct way in table1

  1. Table 2-4. How many repetitions were performed and what was SD for GC area?

Response: We thank the reviewer for the kind comment.

GC analysis performed once with aim of identification.

  1. Table 2-4. Calculations of RI in some cases are not correct. Two compounds with different retention time, cannot have the same RI, for example, RI 1252 both for 10.452 min and 11.356 min (Table 2).

Response: We thank the reviewer for the valuable comment

We revised the data carefully and edited it as suggested by the reviewer.

  1. Which method was used for T.vulgaris essential oil: described in 4.5.1 or in 4.5.3?

Response: We agree with the reviewer

We prepared the essential oil using only one method and we removed the other method that written by mistake from the article as follow

4.5.2. Preparation of Essential oil (Thyme Vulgaris)

The fresh leaves were collected and subjected to hydrodistillation in a Clavenger's apparatus for 5 hrs., according to the procedure described in the Egyptian Pharmacoepia40.

  1. The sample preparation procedure still is not clear: what amount of plant material and solvent was used?

Response: We thank the reviewer for the valuable comment

We demonstrated the method of plant extracts and essential oil preparation in the material and methods section as follow

4.5.1. Preparation of O.europaea leaves extract

The O. europaea leaves were dried at room temperature (26 ± 2 °C) for two weeks and ground to a fine powder. The ground leaves (1 g) were extracted with 10 mL of high purity ethanol, dichloromethane, chloroform and ethyl acetate solvent. A whatman filter paper No. 1 was used to filtrate the extracts, and then a rotary evaporator was used to concentrate the extracts at 40 °C using 50 mL centrifuge tubes. The extracts were dried in a glassy desiccator, then; the residues of the powder were stored for analysis. Extraction was carried out three times with the same volume of solvent added repeatedly.

4.5.2. Preparation of Essential oil (Thyme Vulgaris)

The fresh leaves were collected and subjected to hydrodistillation in a Clavenger's apparatus for 5 hrs., according to the procedure described in the Egyptian Pharmacoepia40.

6.Producers and purity of all chemical compounds should be mentioned in the paper

Response: We thank the reviewer for the valuable comment

We edited it as suggested by the reviewer in the material and methods section as follow

Materials

The T. vulgaris and O. europaea leaves used in this work were obtained from the Medicinal, Aromatic, and Poisonous Plants Experimental Station, Faculty of Pharmacy, Cairo University, and the Department of Medicinal and Aromatic Plants, Faculty of Agriculture, Cairo University, Egypt.

All solvents used were of LC/MS grade purchased from J. T. Baker (The Netherlands).

All other. chemicals and standards were purchased from Sigma Aldrich (St. Louis, MO, USA).

Round 2

Reviewer 1 Report

Some of my suggestions were not taken into consideration

For instance, in the title, the authors only mentioned Olea europaea leaf extract and Thymus vulgaris, but in the manuscript, B. frankincense was also investigated

The disc diffusion assay is no longer acceptable in most journal because it is not a sensitive technique when evaluating the antimicrobial activity.

L26: what is the unit of the values reported here?

L27: “….these compounds….”which compounds are you referring to?

Table 1: n =3. N should be in italics

I still have serious reservation of the identity of compounds in Table 2 and 3. The published RRI and the calculated ones obtained in the current study should be included because the RRI will assist with the correct identification of the compounds. What is % similarity of the compounds listed? For compounds that could not be correctly identify, the authors should rather use “tentative identification”

Table 4: what id p- cineole this is not common

Some compounds are written with symbol and in letter (for instance α-pinene and gamma terpinene), only the symbol should be used

L281: F. oxysporum should be in italics

In 485, it is mentioned that ANOVA was used but in the test no reference on statistical analysis

Author Response

To:  Mr. Norbert-Zsolt Dorner

Assistant Editor, Molecules  Dear reviewerdorner@mdpi.com

Manuscript Title: Antifungal Evaluation and Molecular Docking Studies of Olea europaea leaf extract ,Thymus vulgaris and Boswellia carteri essential oil as Prospective Fungal Inhibitor Candidates

Corresponding Author:

Professor. Soheir N. Abd El-Rahman

Email:    soheirkenawy@yahoo.com                     

Dear Mr. Norbert-Zsolt Dorner,

We are grateful to the editors and reviewers for their time and constructive comments on our manuscript. We think reviewers’ constructive comments greatly improved manuscript. We have carefully addressed all the comments. The corresponding changes and refinements made in the revised paper are summarized in our response below.

Comments from the editors and reviewers:
-Reviewer
Dear reviewer, allow me to appreciate your very valuable and precious comments which considerably helped to improve and strengthen the manuscript. We are honored to offer you the following replies hoping that you will find them satisfactory and properly addressing your concerns. We have addressed the reviewer’s suggestions and revised the manuscript accordingly. Please find attached a detailed point-by-point response to reviewer's concerns.

  1. For instance, in the title, the authors only mentioned Olea europaea leaf extract and Thymus vulgaris, but in the manuscript, B. frankincense was also investigated

Response: We totally agree with the reviewer.

We revised the article and corrected it. We added it to the title as suggested by the reviewer as follow

Antifungal Evaluation and Molecular Docking Studies of Olea europaea leaf extract ,Thymus vulgaris and Boswellia carteri essential oil as Prospective Fungal Inhibitor Candidates

  1. The disc diffusion assay is no longer acceptable in most journal because it is not a sensitive technique when evaluating the antimicrobial activity.

Response: We thank the reviewer for the valuable comment. We agree with the reviewer and we repeated antifungal activity using a food poisoned technique. The new method is more sensitive technique and results are more valuable and acceptable.

We edited it at all manuscript

2.3. Biological Activity

The antifungal properties of the ethanol, dichloromethane, chloroform, and ethyl acetate from O. europaea leaf extracts, and the oils of T. Vulgaris and B. carteri  against F. oxysporum f. sp. lactucae were evaluated in this investigation, as shown in Fig.3 Fig.4  and Table 1. The results revealed that the T. Vulgaris essential oil and the dichloromethane, chloroform and ethanol extracts of O. europaea leaf extract leaves efficiently inhibited the mycelial growth of the F. oxysporum f. sp. lactuca compared to control treatment. Whereas the ethyl acetate extracts did not exhibit antifungal effects at the tested concentration. Data illustrated in Fig. (4) indicate that all the tested plant’s extracts except ethyl acetate extract significantly decreased the mycelial growth of the pathogenic fungi compared to control treatment. The oil of T. Vulgaris showed a maximum inhibition percentage (94.11) at the tested concentration (75. 5 mg/mL), against the fungus.  Moreover, the chloroform, dichloromethane and ethanol extracts exhibited the maximum inhibition (67.058, 64.71 and 50. 59 mm, respectively) at the tested concentration (75. 5 mg/mL) .

. L26: what is the unit of the values reported here?

Response: We thank the reviewer for the valuable comment.

We edited it.

  1. L27: “….these compounds….”which compounds are you referring to?

Response: We thank the reviewer for the kind comment

We referred to compounds mentioned before in the abstract (compounds exerted high activity in molecular modeling). We wrote it in the abstract as follow

The GC-MS analysis revealed that the major and most vital compounds of the T. vulgaris EO, chloroform, and dichloromethane were thymol, carvacrol, tetratriacontane, and palmitic acid. Moreover, molecular modeling revealed the activity of these compounds against F. oxysporum.

  1. Table 1: n =3. N should be in italics

Response: We agree with the reviewer.

we repeated antifungal activity using a food poisoned technique. We changed Table 1 according to results of a new method as follow

Table 1. IC50 of different plant extracts against F.oxysporum

Plant extract

IC50 (mg/ml)

Thyme oil

28.99 (µl/ml)

Ethanol extract

49.60

Dichloromethane extract

32.51

Chloroform extract

31.35

Ethyl acetate extract

521.9

B. carteri

176.2

  1. I still have serious reservation of the identity of compounds in Table 2 and 3. The published RRI and the calculated ones obtained in the current study should be included because the RRI will assist with the correct identification of the compounds. What is % similarity of the compounds listed? For compounds that could not be correctly identify, the authors should rather use “tentative identification”.

Response: We thank the reviewer for the valuable comment.

We agree with the reviewer. We edited tables of GCMS analysis of all samples and added the %similarity of each compound. We carefully revised the RI of each compound by comparing RI with reference RI from published papers to confirm the identification of compounds.

  1. Table 4: what id p- cineole this is not common.

Response: We thank the reviewer for the kind comment.

We confirmed from cineole identification by high value of % similarity (96) and it was previously identified in Thyme oil analysis[1].

  1. Some compounds are written with symbol and in letter (for instance α-pinene and gamma terpinene), only the symbol should be used.

Response: We thank reviewer for the valuable comment.

We edited it as required.

  1. L281: F. oxysporum should be in italics

Response: We thank reviewer for the kind comment.

We edited it.

  1. In 485, it is mentioned that ANOVA was used but in the test no reference on statistical analysis

Response: We thank reviewer for the valuable comment.

We agree with the reviewer. We repeated antifungal activity using a food poisoned technique. We performed statistics and illustrated details of statistics  in manuscript in the materials and method section as follow                 

 4.8 Statistical data interpretation

Data analysis and graphs were made using the GraphPad Prism Version 9 program. The data are expressed as an arithmetic mean, standard deviation and 95 percent confidence interval for the IC50 parameter. IC50 was calculated as a concentration of the tested compound which decreases the mycelial growth by half between the base and maximum. P values less or equal 0.05 were considered statistically significant41.

This manuscript is a resubmission of an earlier submission. The following is a list of the peer review reports and author responses from that submission.

Round 1

Reviewer 1 Report

Although there is a lot of information in this work, there is a major problem with the English that makes it difficult to easily read the manuscript. 

The work is scientifically solid, and the findings are very interesting. The authors should edit the manuscript.

A few comments that might help improving the manuscript:

Line 47: the references should not be separated with “and”

Line 49: The names of species such as Thymus should be written in italics throughout the manuscript

Line 49: Authors write “They recommended…” Who recommended?

Line 216: “These results in agreement with18”. Please describe what were the main conclusions of this reference

Line 245: It would be better not to use the reference at the beginning of the sentence. This has to be edited.

Materials and methods

I am concerned about the fact that the authors present in the results section a relative quantification of the extracts’ composition without mentioning the following: How many replicates of the samples have they studied? How were the results processed statistically? What are the standard deviations of these results? Have the authors used an internal standard?  

Author Response

Dear reviewer1, we as authors appreciate your valuable and precious comments which considerably helped to improve and strengthen the manuscript. We are to provide the following replies hoping you will find them satisfactory and properly addressing your concerns. We have addressed the reviewer’s suggestions and revised the manuscript accordingly. Please find attached a detailed point response to the point of concerns.

Reviewer 2 Report

Major

The manuscript is poorly written and must be edited by and English speaking person. The GC analysis of the T. vulagaris and Boswellia carteri have been previously investigated by several authors. In the title, Boswellia carteri is not mentioned but the oil has been investigated.

Hydrochlorothiazide has not been previously reported as major constituent of the oil, but used as internal standard (Tafest eta l., 2011, Journal of Applied Pharmaceutical Science Vol. 11(01), pp 144-151). I have serious reservation on the identity of the compounds provided in Table 1. The authors should also try to report on these compounds using the common names for better comparison. It is not clear throughout the manuscript if the GC-MS is similar to GC-mass as consistently used by the authors. The authors should use “ % area” in Table 1. In addition, when reporting on GC-MS composition, it is important to use relative retention index (RI) and not retention time. The previous RI should also be included for comparison purposes. The authors should consult with the guidelines to authors before submitting the manuscript. The paper is poorly written and structured, there is no flow and there is no clear reason why the study was conducted. For instance “the aims of this study, to 78 isolated and identify the purified of F. oxysporum forma species Lactucae, using the ITS 79 sequence of the conserved ribosomal DNA, to evaluate the biological activities of some 80 plant extract i.e Thyme (Vulgaris), Boswellia (frankincense) and olive leaves against 81 pathogenic plant fungus. Besides, to determine the active ingredient compounds for 82 these extract was…”this far from what is described in the title of the study. In addition, the sentence does not make sense .., you mentioned that you used EO for T. vulgaris not extract, this is very confusing. The quality of the image in Figure 1 is very poor. It is not clear if the authors is discussing about the EO of t vulgaris of the extract. The major constituent of T vulgaris is thymol and carvacrol. It is not clear why the composition of Boswellia carteri was also not reported. The authors should use symbol such as “α “ant not alpha. I am not convinced about the correct identification displayed in Table 1, 2 and 3. These compounds must be re assessed. And when standards were not used, the authors should use “tentative identification”. The disc diffusion assay is a very old method and no reliable anymore. The antimicrobial activity was investigated without a proper positive control which scientifically is not acceptable. In table 4, the number of replicates should be provided. In table, the letters superscript letters a, b, c and are used, but there is no legend to explain their meaning. I have no doubt that this manuscript was not carefully prepared and edited.

L2: Thymus should be written in full in the title

L2: the authors should either used the scientific name of Olive or common name of T vulgaris to be consistent

L13: Fusarium should be written in full at the beginning of the sentence

L15: beside, use…. This sentence should be rephrased

L16: these extracts (s should be added to extract)

L19: not sure why ethanol and ethyl acetate start with a upper case letter

L60: Ergosterol why always use upper case?

L150: not sure why the solvent start with an upper case

L226: the oils are not extracted but isolated

L227: throughout the manuscript, the authors is using “Gc-mass”, does this refer to GC-MS? The correct terminology should be used.

L229: Hydrochlorothiazide, Thymol, why using upper case for some compounds?

L231: “…..agrees with Moghtader 2012, who reported that GC chromatography the combinations….” What does is the meaning of this sentence

L232 and L237: there is a lack of consistency of some compounds e.g P-cymene and p-cymene

L326: the authors did not take time to proof read the manuscript and there are several careless errors. 60 ̊c and 24Hrs, sometimes, there is no consistency sometime there is a space between number and unit and sometimes no space.

L326 and 330: 24 hrs and 72 hours, where is the consistency

L344 346; no consistency when writing degree celsius

Author Response

Dear reviewer, we thank you for the constructive remarks and comments on the manuscript. We have taken the comments on board to improve and clarify the manuscript. Please find below a detailed point-by-point response to all comments

Reviewer 3 Report

The present study has several tasks, such as , the aims of this study, to isolated and identify the purified of F. oxysporum forma species Lactucae; to evaluate the biological activities of  thyme, boswellia and olive leaves against pathogenic plant fungus; to determine the active ingredient compounds of plant material.

The method part is the weakest part of this paper as some experiments cannot be repeated:

  • proportions of plant material and solvent used for extraction are not clear.
  • how was prepared extract for GC analysis (was it diluted, what solvent, concentration)?
  • what about repeatability of extraction and GC analysis? How many runs were performed?
  • why thymus and boswellia were not extracted in the same way as olive leaves and vise versa?
  • add latin name of olive tree in the manuscript.
  • add manufacturer and purity of used materials and reagents.
  • latin names should be in italic in the text and reference list.
  • add journal to the reference in 459 line
  • also some words  written in capital letters in the middle of the sentence/table or some words with the first capitalized letter where shouldn't be, do not look well...

Author Response

Reviewer3
Dear reviewer, we appreciate your kind concern and hence valuable comments to which considerably helped to improve and strengthen the manuscript. We have tried our best to comply with and fulfill the required points. Hoping the following replies, you find satisfactory and properly addressing your concerns. We have addressed your suggestions and revised the manuscript accordingly. Please find attached a detailed point-by-point response to the reviewer's concerns.

Round 2

Reviewer 2 Report

Although, significant changes have been made, the quality of the paper is still very poor and in my opinion cannot be accepted for publication. The manuscript must be edited by an English speaking person. I still do not understand what is the GC mass? IS this GC-Mass pair for identification of compounds responsible for the activity or GC-MS to identify the EO constituents? The introduction is poorly written and lacks focus. The paragraphs are not linked to each other and in my opinion, it is still not clear why the study was conducted. Olea europaea should be written O. europaea in the text, but the authors are writing this and also T. vulgaris in full. Table 1, 2 and 3 cannot be accepted for publications since the relative retention indexes are not provided. Rather the authors reported on the retention time which is not correct. In addition, most the compounds reported are very problematic and I have the impression that these names have been reported as such. In addition, some retention times have one decimal and other two decimals. There is no positive control in Table 4. The discussion is very poor and should be combined with the results

L18: I still do not understand what is GC Mass (Is this GC mass spectrometry or GC mass?)

L19: F. oxysporum, must be in italics

L21-22: “…. Of the tested” This sentence is incomplete

L24: Thymus vulgaris should be T. vulgaris

L49: Essential oils (EOs) constitutes? Are you talking about constituents?

L53: delete “the”at the beginning of the sentence

L135-137: In addition, dichloromethane Olea europaea leaves extract contained tetra-135 tetracontane (CAS), Pentatriacontane (CAS), 25,26,27-TRISNORCHOLECALCIFER-24-136 AL, Stigmast-5-en-3-ol(3á,24S)(CAS), Ç-Sitosterol, 03027205002 FLAVONE 4'-OH,5-137 OH,7-DI-O-GLUCOSIDE and QUERCETIN. I am sure why these compounds are written in capital letters. The authors have also provided the CAS which is not relevant here. I have the impression that, the constituent obtained by GC-Ms is reported without proper evaluation (comparison of MS and similarity percentage).